# The Impact of Acute Systemic Inflammation Secondary to Oesophagectomy and Anastomotic Leak on Computed Tomography Body Composition Analyses

**DOI:** 10.3390/cancers15092577

**Published:** 2023-04-30

**Authors:** Leo R. Brown, Michael I. Ramage, Ross D. Dolan, Judith Sayers, Nikki Bruce, Lachlan Dick, Sharukh Sami, Donald C. McMillan, Barry J. A. Laird, Stephen J. Wigmore, Richard J. E. Skipworth

**Affiliations:** 1Clinical Surgery, University of Edinburgh, Royal Infirmary of Edinburgh, Edinburgh EH16 4SA, UK; mramage@exseed.ed.ac.uk (M.I.R.);; 2Academic Unit of Surgery, University of Glasgow, Glasgow Royal Infirmary, Glasgow G31 2ER, UK; 3St Columba’s Hospice, Edinburgh EH5 3RW, UK; 4Department of General Surgery, Borders General Hospital, Melrose TD6 9BS, UK; 5Department of General Surgery, Victoria Hospital, Kirkcaldy KY2 5AH, UK; 6Department of General Surgery, Dumfries and Galloway Royal Infirmary, Dumfries DG2 8RX, UK; 7Institute of Genetics and Molecular Medicine, University of Edinburgh, Edinburgh EH4 2XR, UK

**Keywords:** systemic inflammation, anastomotic leak, oesophagectomy, computed tomography, body composition, cancer

## Abstract

**Simple Summary:**

Measures of body composition have been used extensively for prognostication across an array of malignant and benign diseases. Systemic inflammation is both a key driver of cancer cachexia and a common finding in patients presenting with acute pathology. However, its influence on estimates of body composition remains poorly understood. Postoperative anastomotic leak represents a relatively unique opportunity to model the effects of acute, severe systemic inflammation on body composition. This study found that systemic inflammation has a marked effect on CT-derived estimates of body composition. Decreased quantities of skeletal muscle and increased measures of intramuscular and subcutaneous adipose were observed following the inflammatory insult. Radiodensity across muscle and adipose tissues trended towards that of water, likely secondary to oedema. Future research utilising body composition should be interpreted with consideration of the potential of influence of underlying inflammatory status.

**Abstract:**

This study aimed to longitudinally assess CT body composition analyses in patients who experienced anastomotic leak post-oesophagectomy. Consecutive patients, between 1 January 2012 and 1 January 2022 were identified from a prospectively maintained database. Changes in computed tomography (CT) body composition at the third lumbar vertebral level (remote from the site of complication) were assessed across four time points where available: staging, pre-operative/post-neoadjuvant treatment, post-leak, and late follow-up. A total of 20 patients (median 65 years, 90% male) were included, with a total of 66 computed tomography (CT) scans analysed. Of these, 16 underwent neoadjuvant chemo(radio)therapy prior to oesophagectomy. Skeletal muscle index (SMI) was significantly reduced following neoadjuvant treatment (*p* < 0.001). Following the inflammatory response associated with surgery and anastomotic leak, a decrease in SMI (mean difference: −4.23 cm^2^/m^2^, *p* < 0.001) was noted. Estimates of intramuscular and subcutaneous adipose tissue quantity conversely increased (both *p* < 0.001). Skeletal muscle density fell (mean difference: −5.42 HU, *p* = 0.049) while visceral and subcutaneous fat density were higher following anastomotic leak. Thus, all tissues trended towards the radiodensity of water. Although tissue radiodensity and subcutaneous fat area normalised on late follow-up scans, skeletal muscle index remained below pre-treatment levels.

## 1. Introduction

Oesophagectomy remains the cornerstone of curative treatment for patients with oesophageal cancer. While outcomes have improved markedly over recent decades [1], the procedure is still associated with high levels of morbidity and mortality. Contemporary estimates regarding the incidence of anastomotic leak following oesophagectomy range between 10–20% internationally [2]. Although certainly associated with a prolonged inpatient and critical care stay, the influence of anastomotic leak on postoperative mortality and long-term survival is less clear [3,4].

Computed tomography (CT) body composition analysis has been used extensively across malignant and benign surgical populations, both in the diagnosis of cachexia or sarcopenia and, more broadly, for prognostication. A low skeletal muscle index (SMI), indicative of sarcopenia, is associated with decreased survival in patients with incurable oesophageal cancer [5]. In those with resectable disease, it is predictive of postoperative complications, disease-free survival, and overall survival [6]. For patients undergoing an emergency laparotomy, necessitated by benign or malignant pathology, the presence of sarcopenia has similarly been linked to adverse outcomes [7,8]. Low skeletal muscle density, thought to represent myosteatosis (infiltration of fat into skeletal muscle) is also prognostic across these patient cohorts, both alone [7,9] and in combination with SMI [10]. A rapidly growing body of research has sought to explore the influence of sarcopenia across countless disease types and clinical scenarios [11]. The negative implications of low muscularity have been consistently evident.

Elevated markers of systemic inflammation are known to be associated with decreased muscle mass and density in patients with cancer [12]. Indeed, the chronic inflammatory response forms a central tenet of the complex pathophysiological processes underpinning cancer cachexia. Further systemic inflammation is evident following major surgery, including cancer resections, particularly via an open-operative approach [13,14]. However, immediate post-operative CT scans that could demonstrate the impact of such systemic inflammation are not carried out routinely in clinical practice. Postoperative complications, such as anastomotic leak, are associated with a further severe systemic inflammatory response [15], and afford the opportunity for early cross-sectional imaging, thus providing insight into the effects of severe systemic inflammation on body composition.

This study aimed to longitudinally assess the impact of a systemic inflammatory response on CT body composition analyses in patients who have undergone oesophagectomy and experienced an anastomotic leak. The authors hypothesised that the influence of tissue oedema, associated with this severe inflammatory response, may influence CT-body composition estimates.

## 2. Materials and Methods

Data were collected for consecutive patients who experienced an anastomotic leak following a two-stage transthoracic (Ivor-Lewis) oesophagectomy over a ten-year period, between January 2012 and January 2022, at a tertiary referral centre. All patients were discussed at a multidisciplinary team meeting and underwent treatment with curative intent. Surgical resection was performed alone, or following neoadjuvant chemo(radio)therapy, for adenocarcinoma or squamous cell carcinoma of the oesophagus or gastro-oesophageal junction. Patients were identified from a contemporaneously maintained national database with additional variables sought via electronic patient records.

### 2.1. Staging and Treatment Protocols

All patients were staged prior to commencing treatment via upper gastrointestinal endoscopy (with tissue biopsies) and thoraco-abdominal CT scan. As the patients in this cohort were all planned for treatment with curative intent, they underwent an additional positron emission tomography (PET) CT scan. Select patients had their disease further evaluated using endoscopic ultrasound (+/− fine needle aspiration) and/or laparoscopy to complete pre-treatment staging.

Patients with resectable but locally advanced tumours (cT3+), or evidence of nodal disease, routinely underwent neoadjuvant chemo(radio)therapy prior to surgical resection, unless contraindicated. Neoadjuvant treatment regimens varied over the study period, as determined by the standard of care at the time. The operative management for all included patients was a two-stage transthoracic oesophagectomy, via an open approach across both abdominal and thoracic components.

Patients who were suspected to have experienced an anastomotic leak underwent an urgent CT scan of the chest and abdomen. This investigation was prompted by postoperative clinical symptoms and signs, or biochemical markers suggestive of sepsis.

### 2.2. Anastomotic Leak Definition

As per the Esophagectomy Complications Consensus Group, an anastomotic leak was defined as a full-thickness gastrointestinal defect, involving the anastomosis, staple line, or conduit, irrespective of presentation or method of intervention [16]. Anastomotic leaks were graded as follows:Type I: Localised defect requiring no altered therapy/treated medically or with dietary modification only.Type II: Localised defect requiring interventional but not surgical therapy, for example, interventional radiology drain, stent or bedside opening, and packing of incision.Type III: Localised defect requiring surgical therapy.

### 2.3. Other Definitions

Cancer cachexia was diagnosed and classified as per Fearon et al.’s consensus definition [17]. In short, this requires unintentional weight loss of >5%, or greater than 2% in patients with a body mass index (BMI) < 20 or evidence of sarcopenia for a diagnosis of cachexia. BMI was classified as underweight (<18.5 kg/m^2^), healthy (18.5–24.9 kg/m^2^), overweight (25.0–29.93 kg/m^2^), or obese (≥30 kg/m^2^) [18]. The Malnutrition Universal Screening Tool (MUST) score was calculated as per the British Association of Parenteral and Enteral Nutrition (BAPEN) guidance [19]. Comorbidity was described using the Charlson Comorbidity Index [20] and the American Society of Anaesthesiologists (ASA) grading systems [21].

The presence of residual tumour (R classification) was defined as per the Royal College of Pathologists guidance, with circumferential microscopic margins considered positive (R1) when tumour is within 1 mm of the cut margin [22].

### 2.4. Systemic Inflammation

Haematological and biochemical results were reviewed at three time points via electronic patient records: at staging assessment (prior to any neoadjuvant treatment), following completion of neoadjuvant treatment (pre-operatively), and on the day of the anastomotic leak diagnosis. The neutrophil-lymphocyte ratio (NLR) was calculated by dividing the neutrophil count by the lymphocyte count. C-reaction protein (CRP) was not routinely sampled at staging or pre-operative appointments, during the study period, but was monitored during the postoperative course and thus was available at the time of anastomotic leak.

### 2.5. Computed Tomography (CT) Body Composition

Staging and early postoperative, portal-venous phase CT scans, performed to investigate a potential anastomotic leak, were analysed for each included patient. If either staging or early postoperative CT scans were not available, patients were excluded from analyses. In patients who underwent neo-adjuvant anti-cancer therapy, additional pre-operative scans were also examined. Where available, the earliest follow-up CT scan, completed >6 months following surgical resection, was also retrieved to assess the reversibility of changes noted on postoperative scans. If the formal report from the radiologist, or other investigation findings (such as endoscopy) at that time point, indicated recurrent disease, these follow-up scans were excluded. This was due to the possibility that any observed altered body composition may be secondary to the known changes associated with recurrent malignancy. In all cases, contrast-enhanced portal venous phase imaging was utilised for analysis.

CT body composition was performed using the Data Analysis Facilitation Suite (DAFS) by Voronoi Health Analytics Ltd. (Voronoi Health Analytics, Vancouver, Canada, 2021, https://www.voronoihealthanalytics.com (Accessed on 1 December 2022)). DAFS uses non-linear image processing algorithms to provide multi-slice segmentation of tissues and automated vertebral level annotation across axial slices (Figure 1). Validation against manual segmentation analysis has confirmed that a high level of accuracy is achieved by these segmentation algorithms [23]. Cross-sectional area (cm^2^) at the mid-3rd lumbar vertebral level (L3), was measured for skeletal muscle, visceral adipose tissue, subcutaneous adipose tissue, and intramuscular adipose tissue. The cross-sectional area of muscle was normalised for height (m^2^) to create a skeletal muscle index (cm^2^/m^2^). Skeletal muscle density (HU) was measured as a mean across the same region of interest. The L3 axial level was chosen owing to its strong association with whole-body estimates of body composition [24]. Furthermore, as all anastomoses were sited intra-thoracically, this level was remote from the site of postoperative complication.

### 2.6. Comparison of Anterior and Posterior Muscle Groups

All patients were scanned in the supine position and thus oedema would be expected to be more evident in posterior tissues. Automated segmentation was augmented manually to isolate individual muscle groups across the L3 vertebral level. Measurements for the cross-sectional area and radiodensity of rectus abdominus (anteriorly) were compared to multifidus and erector spinae (posteriorly).

### 2.7. Statistical Analysis

Data analysis was performed using R 4.2.2 (R Foundation for Statistical Computing, Vienna, Austria) with packages including *tidyverse*, *ggplot2*, and *finalfit*. Categorical data were summarised using frequencies and percentages. Continuous variables were presented using either the median and interquartile range (IQR) or mean and standard deviation (SD) based on visual and statistical evaluation for normality. Differences between continuous, normally distributed variables, on repeated measure, were determined by a paired, two-sided *t*-test or repeated measures ANOVA if considering more than two time points. Two-tailed *p* values of <0.050 were deemed statistically significant.

## 3. Results

Overall, 25 patients experienced an anastomotic leak following two-stage, transthoracic oesophagectomy during the 10 year study period (Figure 2). Five of these patients were excluded: two due to a lack of postoperative CT scans, two due to poor quality imaging precluding accurate analysis, and one due to the resection having been performed for a non-malignant (caustic) aetiology. This resulted in a cohort of 20 patients (18 male) with a median age of 65 years (range 47–73, Table 1). Notable comorbidity was present at baseline, with only 20% of patients being classified as American Society of Anaesthesiologists (ASA) grade one, and 45% having a Charlson Comorbidity Index score >5. The median BMI amongst the cohort was 27.3 kg/m2 (range 19.2–40.9) with only 30% of the cohort being of “healthy” weight. A total of 10 patients had reported weight loss prior to their diagnosis of oesophageal cancer. Amongst those who had lost weight, eight patients (40% of the overall cohort) met Fearon et al.’s diagnostic criteria for cachexia prior to commencing treatment [17]. Most tumours were situated in the lower oesophagus (65%) and were locally advanced (>cT3) or had evidence of nodal disease at the point of clinical staging (both 85%). Histology was reported as adenocarcinoma for 16 patients and squamous cell carcinoma for the remaining 4.

### 3.1. Neoadjuvant Chemotherapy

A total of 15 patients underwent neoadjuvant chemotherapy (75%) prior to surgical resection (Appendix A, Table A1). The majority had Cisplatin/5-Fluorouracil (n = 13) while two contemporary patients had FLOT (Fluorouracil, Leucovorin, Oxaliplatin, and Docetaxel) combination chemotherapy. One patient, who had a squamous cell carcinoma, had neoadjuvant chemoradiotherapy as per the CROSS (Carboplatin, Paclitaxel, and radiotherapy) regimen. Only one patient did not complete their planned course of chemotherapy, owing to experiencing new claudication after one cycle. No other toxicities or complications were reported during neoadjuvant treatment.

Weight was stable across neoadjuvant treatment (*p* = 0.468). Neutrophil-lymphocyte ratios, as a marker of systemic inflammation, were also comparable before and after treatment (*p* = 0.892). On CT analysis, the skeletal muscle index was decreased after neoadjuvant treatment (mean difference −3.58 cm^2^/m^2^, *p* < 0.001, Appendix B, Table A2). Skeletal muscle density (*p* = 0.283) was unchanged. The mean cross-sectional area and density of visceral, subcutaneous, and intramuscular fat were also comparable.

### 3.2. Surgical Resection and Anastomotic Leak

All included patients underwent a two-stage, transthoracic oesophagectomy via an open approach across both phases. Most patients had a poorly differentiated tumour (85%) on histology and pathological staging exceeded clinical staging in 45% of patients. Seven patients had an R1 resection. The median postoperative inpatient stay was 41.5 (IQR: 32.5–55.5) days.

An anastomotic leak was diagnosed at a median of 8.5 days (IQR: 7–11) following surgery. Most of these leaks were “type 1” (n = 12) requiring only conservative management. Three leaks were “type 2”, necessitating radiological drainage, and five required a re-look thoracotomy +/− laparotomy. At the point of diagnosis, high levels of systemic inflammation were evident with a mean CRP of 227 (SD: 100) and a neutrophil-lymphocyte ratio that rose significantly from a mean of 3.26 (SD: 1.38), at staging, to a mean of 17.59 (SD: 5.60) (*p* < 0.001) post-leak (Appendix C, Table A3).

The skeletal muscle area (mean difference: −13.28, *p* = 0.001) and index (mean difference: −4.23, *p* < 0.001) decreased following the inflammatory insult of surgery and subsequent anastomotic leak (Table 2). Mean skeletal muscle density fell from 35.63 (SD 9.84), prior to surgery, to 30.21 (SD 8.29) on the post-leak CT scan. An increased cross-sectional area of intramuscular adipose tissue was also evident (*p* < 0.001). The area of visceral adipose tissue (mean difference: −38.16, *p* = 0.002) decreased, while density rose (mean difference: 13.19, *p* < 0.001) on repeated imaging. A significant increase in the cross-sectional area of subcutaneous fat, across the L3 vertebral level, was conversely seen on comparison of pre-operative (mean: 193.61) and post-leak (mean: 257.12) CT scans (*p* < 0.001).

### 3.3. Recovery of CT Body Composition following Anastomotic Leak

Half of the cohort (n = 10) had a subsequent follow-up CT scan, performed >6 months later (median: 12, range 7.2–47.7 months) with no signs of recurrent malignancy (Appendix D, Table A4). While skeletal muscle density normalised on repeat imaging, muscle index remained below the levels recorded pre-operatively (Figure 3 and Figure 4). Subcutaneous fat area decreased significantly (leak: mean 289.19 cm^2^ vs. follow-up: mean = 170.73 cm^2^, *p* = 0.001), thus returning to a measurement similar to that seen on pre-operative imaging (mean: 193.61 cm^2^).

### 3.4. Comparison of Changes in Anterior and Posterior Muscle Groups

Manual segmentation of rectus abdominus and paraspinal muscles allowed comparison between anterior and posterior muscle groups *(*Table 3 and Figure 5*)*. The cross-sectional area remained stable across skeletal muscle anteriorly (*p* = 0.185), where an observed decrease was noted in posterior muscle groups (mean difference = −3.71 cm^2^/m^2^, *p* < 0.001). Skeletal muscle density was significantly reduced both anteriorly (*p* = 0.020) and posteriorly (*p* < 0.001) on post-leak scans; however, the difference was greater in posterior muscles (−12.27 vs. −6.64 HU).

## 4. Discussion

The present study has demonstrated the effect of acute systemic inflammation, secondary to anastomotic leak after oesophagectomy, on CT-derived estimates of body composition. Longitudinal imaging, surrounding the occurrence of a severe inflammatory stimulus, has highlighted a decreased quantity of skeletal muscle with increased intramuscular and subcutaneous adipose tissue. Adipose tissue radiodensity was significantly higher following the inflammatory stimulus, while skeletal muscle density conversely fell.

Within this cohort, many patients had a significant degree of comorbidity at diagnosis. The proportion of patients who were ASA grade 3 was higher than that reported by other larger-scale studies of United Kingdom oesophageal cancer populations [1,25]. This may suggest they were at greater risk of postoperative complications, including anastomotic leak. Furthermore, almost half of the included patients met the diagnostic criteria for cachexia at the point of clinical staging [17]. There is, at present, a paucity of studies that consider the influence of cachexia on rates of postoperative complications, such as anastomotic leak, with no association having previously been reported [26,27]. Further high-quality evaluation of host risk factors for adverse postoperative outcomes is certainly required. Rates of advanced disease stage (T3 or node-positive) and positive resection margins also appeared elevated in this anastomotic leak cohort [2,25]. This is likely to be suggestive of a more technically challenging surgical resection. While comorbidity is well recognised as a risk factor for anastomotic leak, an increased pathological stage does not appear to be associated according to the existing literature [28]. Skeletal muscle wasting was observed following neoadjuvant chemo(radio)therapy in this study cohort, in keeping with a recent meta-analysis, which concluded that muscle mass significantly decreases during neoadjuvant treatments [29].

To the best of our knowledge, this is the first study to longitudinally evaluate the impact of an acute severe systemic inflammatory response on CT-derived estimates of body composition. Specifically, our results demonstrate decreased skeletal muscle measurements following surgery and anastomotic leak. Estimates of subcutaneous and intramuscular fat were conversely higher following the inflammatory insult. While the amount of visceral fat decreased, this change is likely to reflect the greater omentum being moved into the thoracic cavity alongside the gastric conduit during oesophagectomy. The differing measurements observed, across a short timeframe, raise the possibility that acute systemic inflammation may influence CT-derived body composition analysis. It is difficult to ascertain, however, whether the observed changes to body composition are real or apparent. Marked metabolic changes are known to occur in response to injury, with hypermetabolism and increased protein catabolism both seen in critically-ill patients [30]. Rapid depletion of the body’s protein stores, primarily from skeletal muscle, have similarly been observed following trauma [31]. However, this may only represent only a proportion of the observed decline in muscle mass. The potential remains for severe inflammation to augment tissue segmentation algorithms. Such findings could have particular implications for studies that have evaluated CT body composition as a prognostic marker in the context of acute inflammatory pathologies. For example, in the context of emergency laparotomy, a large number of patients will have experienced a significant inflammatory insult prior to undergoing their diagnostic CT scan. It is, therefore, possible that acute systemic inflammation may result in a proportion of patients being mis-labelled as having sarcopenia or myosteatosis [7,8].

Tissue oedema is a frequently observed sequela of critical illness and major surgery, including oesophagectomy [32]. Inflammatory stimuli promote a loss of endothelial barrier function resulting in increased permeability and tissue oedema [33]. Additionally, the secretion of hormones, such as arginine vasopressin, and the activation of the renin-angiotensin-aldosterone system can promote renal salt and water retention following trauma or surgery. We would postulate, therefore, that these changes observed in body composition analyses could in part be secondary to tissue oedema. The apparent loss of skeletal muscle may reflect a decreased ability of CT analysis software to identify some areas of muscle, owing to their increasingly oedematous state. Higher quantities of intramuscular adipose tissue were noted following the inflammatory stimulus, and it may be the case that some oedematous muscle is even being labelled as intramuscular fat owing to decreased radiodensity. The possibility of these changes being related to oedema is further supported by the manual segmented comparison of anterior and posterior muscle groups, where the posterior (gravity-dependent) muscle mass was depleted, and anterior muscle mass remained preserved. When compared to standard reference Hounsfield Unit (HU) ranges, a trend towards the radiodensity of water (0 HU) was evident across adipose tissue (−190 to −30 HU) and skeletal muscle (−29 to +150 HU). This is similarly in keeping with tissue oedema. An alternative explanation for the observed decrease in muscle radiodensity is an altered distribution of intravenous contrast in the context of systemic inflammation. Cardiovascular dysfunction can result from septic shock, and this may result in differences between “non-inflammatory” and “inflammatory” scans despite careful contrast gating. Furthermore, severe inflammation can influence tissue microvasculature [34], which may affect the uptake of contrast media and thus their radiodensity.

Amongst the subgroup of patients who had a (recurrence-free) follow-up scan available, estimates of muscle and fat radiodensity returned to levels comparable with their baseline staging CT scan. While the quantity of subcutaneous fat also normalised following resolution of the inflammatory response, the skeletal muscle measurements remained lower. This may be indicative of ongoing dynamic muscle wasting following oesophagectomy, even in the context of an apparently curative cancer resection. Such changes would be in keeping with the existing literature regarding the impaired nutritional status often seen following oesophagectomy [35].

This cohort of patients, who had undergone a major surgical resection and subsequent complication, was chosen to address the research question owing to the routine availability of sequential imaging surrounding a severe inflammatory insult. Few clinical scenarios could readily provide such a comparison across a narrow timeframe, and this should be viewed as a strength of the present study’s design. In most instances where a patient has experienced such acute inflammation, their presentation would be emergent, and thus a “pre-inflammatory” scan, within a close timeframe, is unlikely to be available. By the nature of this scenario, a comparator cohort was not readily available as patients do not routinely undergo post-operative cross-sectional imaging unless complications, such as anastomotic leak, are suspected. As such, it was not possible to draw any conclusions regarding how body composition changes differ between patients who have experienced an uncomplicated post-operative course and those who suffered the additional inflammatory insult of the anastomotic leak. As post-oesophagectomy anastomotic leak represents a complication that is relatively remote from the third lumbar (L3) vertebral level, this minimises the influence of local tissue changes. Furthermore, L3 is the most commonly studied vertebral landmark owing to a good correlation with estimates of whole-body changes in patients with cancer [24]. One limitation of the study is the small sample size. Although a relatively long study period of 10 years was considered, the favourably low rate of anastomotic leak following oesophagectomy, meant that only a finite number of patients could be identified. Fluid balance status was also not reliably recorded in electronic records (where paper notes have been destroyed) and thus cannot be considered alongside body composition analysis. Such information may have been helpful and future studies should aim to prospectively capture this.

## 5. Conclusions

CT-derived estimates of body composition should be interpreted with caution in the context of acute severe systemic inflammation. In particular, decreased measures of the amount and density of skeletal muscle appear to be evident following the inflammatory stimulus of major surgery and subsequent anastomotic leak. Future research utilising body composition variables should be interpreted with consideration of the potential of influence of underlying inflammatory status.

## Figures and Tables

**Figure 1 cancers-15-02577-f001:**
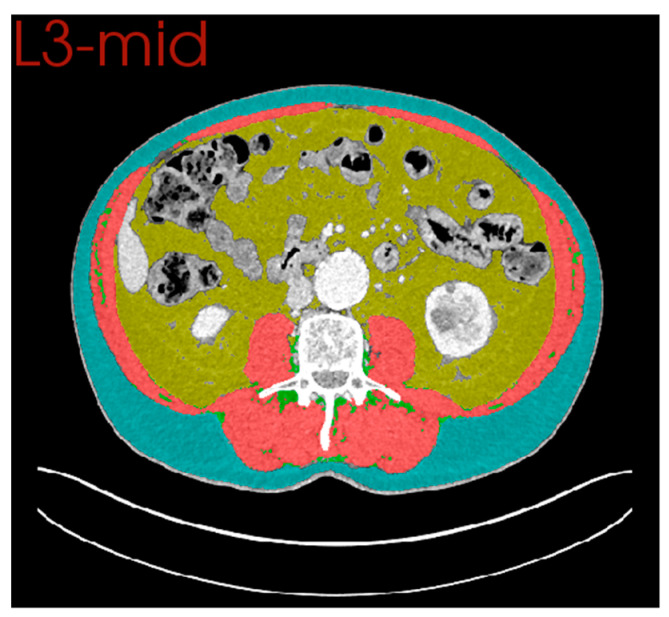
Example of Automated Tissue Segmentation at the L3 Vertebral Level. Blue: Subcutaneous Adipose Tissue. Yellow: Visceral Adipose Tissue. Red: Skeletal Muscle. Green: Intramuscular Adipose Tissue.

**Figure 2 cancers-15-02577-f002:**
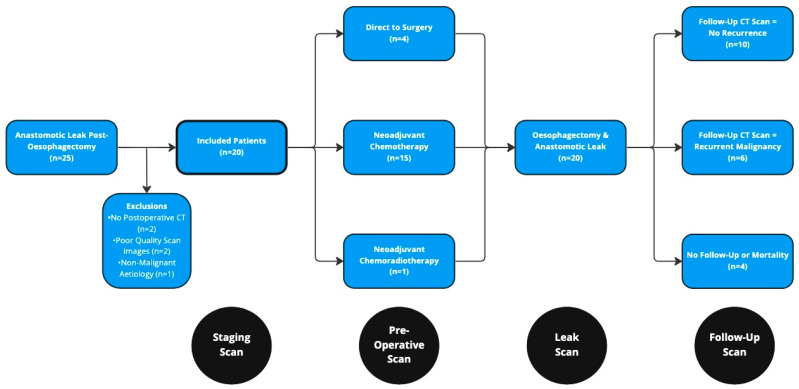
Study Flow Diagram.

**Figure 3 cancers-15-02577-f003:**
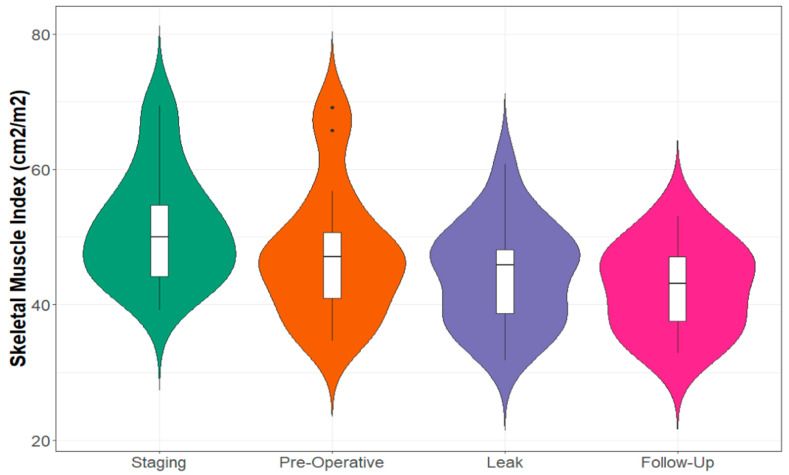
Skeletal Muscle Index at Defined CT Scan Time Points.

**Figure 4 cancers-15-02577-f004:**
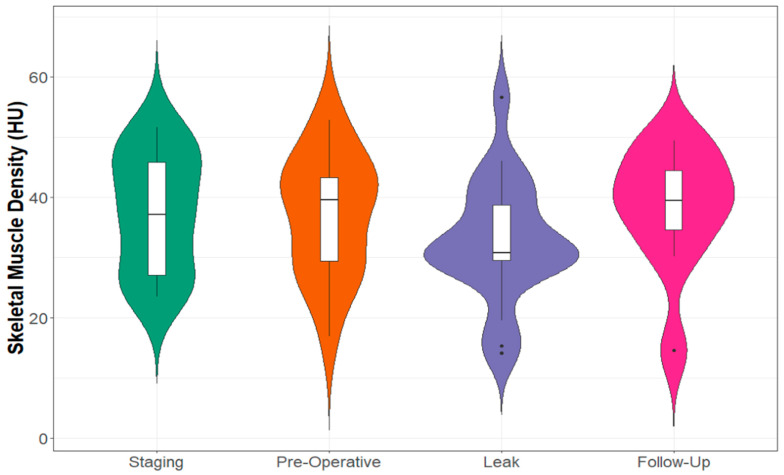
Skeletal Muscle Density at Defined CT Scan Time Points.

**Figure 5 cancers-15-02577-f005:**
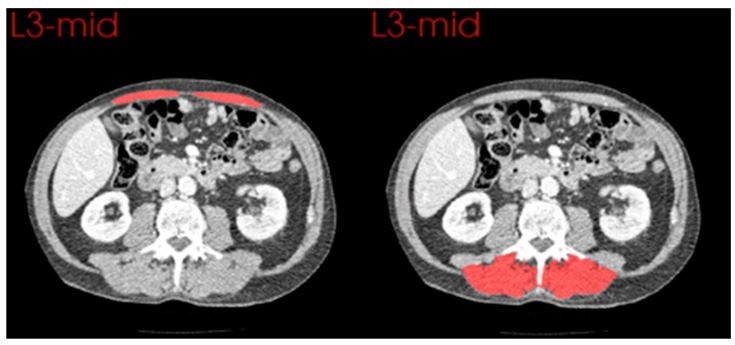
Examples of Manual Segmentation of Anterior and Posterior Muscle Groups. Manually segmented muscle groups highlighted in red (anterior = left, posterior = right).

**Table 1 cancers-15-02577-t001:** Patient and Disease Characteristics.

Age (Years)	Median [IQR]	65 [56–71]
Sex	Male	18 (90%)
	Female	2 (10%)
ASA Grade	1	4 (20%)
	2	8 (40%)
	3	8 (40%)
Charlson Comorbidity Index	0–1	2 (10%)
	2–4	9 (45%)
	>5	9 (45%)
Weight at Diagnosis (kg)	Median [IQR]	84.5 [73–97]
Body Mass Index (kg/m^2^)	Median [IQR]	27.3 [24.5–31.8]
MUST Score	Low Risk	5 (25%)
	Medium Risk	5 (25%)
	High Risk	10 (50%)
Pre-Treatment Weight Loss (kg) *	Median [IQR]	1 [0–4.5]
Pre-Treatment Cachexia	Yes	8 (40%)
	No	12 (60%)
Tumour Site	Middle Oesophagus	2 (10%)
	Lower Oesophagus	13 (65%)
	Gastro-Oesophageal Junction	5 (25%)
Histology	Adenocarcinoma	16 (80%)
	Squamous Cell Carcinoma	4 (20%)
Clinical Tumour Stage	cT1	1 (5%)
	cT2	2 (10%)
	cT3	17 (85%)
Clinical Nodal Stage	cN0	3 (15%)
	cN1	8 (40%)
	cN2	7 (35%)
	cN3	2 (10%)

All data displayed as “number (percentage)” unless stated otherwise. * Documented or self-reported unintentional weight loss during the preceding 6 months. ASA = American Society of Anaesthesiologists. MUST = Malnutrition Universal Screening Tool.

**Table 2 cancers-15-02577-t002:** Effect of an Anastomotic Leak on CT Body Composition.

CT Measurement	Pre-Op Scan Mean (SD)	Leak Scan Mean (SD)	Mean Difference (95% CI)	Percentage Difference	*p* Value
Skeletal Muscle Area (cm^2^)	147.17 (31.38)	133.89 (24.99)	−13.28 (−20.53, −6.02)	9.03%	0.001
Skeletal Muscle Index (cm^2^/m^2^)	48.56 (8.98)	44.33 (7.62)	−4.23 (−6.49, −1.97)	8.71%	<0.001
Skeletal Muscle Density (HU)	35.63 (9.84)	30.21 (8.29)	−5.42 (0.01, 10.83)	15.21%	0.049
Visceral Fat Area (cm^2^)	201.29 (125.48)	163.13 (92.08)	−38.16 (−60.55, −15.77)	18.96%	0.002
Visceral Fat Density (HU)	−100.04 (7.69)	−86.85 (10.41)	13.19 (8.56, 17.82)	13.18%	<0.001
Subcutaneous Fat Area (cm^2^)	193.61 (91.85)	257.12 (109.82)	63.51 (46.22, 80.80)	32.80%	<0.001
Subcutaneous Fat Density (HU)	−103.45 (7.50)	−71.78 (16.08)	29.66 (20.90, 38.43)	28.67%	<0.001
Intramuscular Fat Area (cm^2^)	17.23 (7.15)	20.48 (7.33)	3.24 (1.60, 4.88)	18.80%	<0.001
Intramuscular Fat Density (HU)	−54.22 (6.44)	−49.36 (5.79)	4.85 (0.97, 8.74)	8.95%	0.017

CT: Computed Tomography. HU: Hounsfield Units. Comparison between post-leak CT scan and pre-operative (post-neoadjuvant) CT scan or staging scan in patients who did not undergo neoadjuvant treatment.

**Table 3 cancers-15-02577-t003:** Comparison of Anterior and Posterior Skeletal Muscle.

	Pre-Op Scan Mean (SD)	Leak Scan Mean (SD)	Mean Difference (95% CI)	Percentage Difference	*p* Value
Anterior Muscle					
Skeletal Muscle Area (cm^2^)	12.19 (4.12)	11.13 (4.33)	−1.06 (−2.67, 0.55)	8.70%	0.185
Skeletal Muscle Density (HU)	38.98 (9.59)	32.35 (9.02)	−6.64 (−1.17, −12.10)	16.64%	0.020
Posterior Muscle					
Skeletal Muscle Area (cm^2^)	48.49 (8.26)	44.78 (5.82)	−3.71 (−5.39, −2.03)	7.65%	<0.001
Skeletal Muscle Density (HU)	26.44 (15.63)	15.63 (10.84)	−12.27 (−17.73, −6.81)	46.41%	<0.001

HU: Hounsfield Units.

## Data Availability

The datasets generated and/or analyzed during the current study are not publicly available considering the data are linked to a vulnerable patient population but are available from the corresponding author on reasonable request.

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
