# Peer review of "The Impact of Acute Systemic Inflammation Secondary to Oesophagectomy and Anastomotic Leak on Computed Tomography Body Composition Analyses"

_cancers, 2023, doi:10.3390/cancers15092577_

Round 1
Reviewer 1 Report (Previous Reviewer 1)
Thank you for making suggested revisions.
Reviewer 2 Report (Previous Reviewer 2)
No further comments, the authors addressed most comments.
This manuscript is a resubmission of an earlier submission. The following is a list of the peer review reports and author responses from that submission.
Round 1
Reviewer 1 Report
Your manuscript is of considerable importance in the volume of research evidence relative to the prognostic significance of body composition analysis following cancer resection.
This is an extremely well written, interesting and enjoyable paper to read. There would only be a small number of changes I would suggest to improve the readability:
1. Abstract, line 29, abbreviation for CT needs to be explained in the first instance
2. Include reference for BMI categories, line 124
3. MUST score is mentioned in table 1, yet not mentioned in methods. Suggest including this in section 2.3 of methods, other definitions.
4. 1Charlson Comorbidity Index and American Society of Anaesthesiologists (ASA) grade are mentioned in table 1. Yet not mentioned in methods. Suggest including in methods
5. Section 3.3, line 238, Please clarify what was the mean follow up time for scans (n=10) It is mentioned that the earliest follow up CT scan performed >6months following surgical resection were included, I think it would be helpful to include the median follow up timeframe for these scans.
6. Figures 3 and 4, relate to my previous comment- the title of these figures are skeletal muscle index over time- what timeframe is this referring to?
7. Under section 2.5 CT body composition, possibly consider including image thickness
Author Response
Your manuscript is of considerable importance in the volume of research evidence relative to the prognostic significance of body composition analysis following cancer resection.
This is an extremely well written, interesting and enjoyable paper to read.
Thank you very much for your kind comments. We are very pleased to hear that you enjoyed the paper and really appreciate your feedback.
Individual comments were addressed as below:
Abstract, line 29, abbreviation for CT needs to be explained in the first instance
Corrected
Include reference for BMI categories, line 124
Corrected
MUST score is mentioned in table 1, yet not mentioned in methods. Suggest including this in section 2.3 of methods, other definitions.
Corrected
Charlson Comorbidity Index and American Society of Anaesthesiologists (ASA) grade are mentioned in table 1. Yet not mentioned in methods. Suggest including in methods
Corrected
Section 3.3, line 238, Please clarify what was the mean follow up time for scans (n=10) It is mentioned that the earliest follow up CT scan performed >6months following surgical resection were included, I think it would be helpful to include the median follow up timeframe for these scans.
Corrected
Figures 3 and 4, relate to my previous comment- the title of these figures are skeletal muscle index over time- what timeframe is this referring to?
Amended to hopefully clarify these defined CT timepoints
Under section 2.5 CT body composition, possibly consider including image thickness
Unfortunately we haven't been able to retrieve this information for the cohort. Many scans were performed locally rather than at our hospital (as a tertiary referral centre).
Reviewer 2 Report
This is a study with prospectively collected data with the interesting scope to evaluate the relation between acute severe systemic inflammatory response and CT-derived features of body composition. The findings are logical, thus somewhat expected (ex. decreased skeletal muscle after anastomotic leak and at long-term follow-up, etc). The originality is the scientific documentation of such results for the first time, and this could serve as a starting reference for future research in the field.
The main drawbacks of the study include 1) missing data concerning the whole series of the patients. How many had surgery but not a leak? in case the authors posses this data, it would be nice to present it 2) the results are exclusively limited in the group of patients who developed a leak. Increase of body and fat oedema and a decrease in muscle density are expected as well in operated pts without any leak, due to surgical stress, postop hypercatabolic state, less calory intake, etc. Thus, an interesting clinical question seems to have been missed and it regards the comparison between the drop of body muscle or increase of fat oedema in operated with a leak vs those without leak. Could the authors comment, or could the authors provide any other suggestion about what is the usefulness of their study from the clinical point of view? Another question could be, is there an association of the postop CT findings to leak severity (ex. needing surgery/intervention vs conservative measures?
Nevertheless, the study is well written, stats are sound, methodology could be ok except the drawback discussed above (non-comparison vs non-leak).
Author Response
This is a study with prospectively collected data with the interesting scope to evaluate the relation between acute severe systemic inflammatory response and CT-derived features of body composition. The findings are logical, thus somewhat expected (ex. decreased skeletal muscle after anastomotic leak and at long-term follow-up, etc). The originality is the scientific documentation of such results for the first time, and this could serve as a starting reference for future research in the field.
Thank you very much for your positive comments regarding the paper. We really appreciate your feedback.
The main drawbacks of the study include 1) missing data concerning the whole series of the patients. How many had surgery but not a leak? in case the authors posses this data, it would be nice to present it 2) the results are exclusively limited in the group of patients who developed a leak. Increase of body and fat oedema and a decrease in muscle density are expected as well in operated pts without any leak, due to surgical stress, postop hypercatabolic state, less calory intake, etc. Thus, an interesting clinical question seems to have been missed and it regards the comparison between the drop of body muscle or increase of fat oedema in operated with a leak vs those without leak. Could the authors comment, or could the authors provide any other suggestion about what is the usefulness of their study from the clinical point of view?
Thank you for your comments - we agree that this it is likely that a similar, yet less pronounced, effect would be seen amongst the cohort who did not have a postoperative leak. We did discuss the possibility of comparison with a matched cohort as an authorship group. Unfortunately, as patients in our service do not routinely undergo a postoperative CT scan, it would be difficult to compile a true comparator cohort. Post operative scans are usually prompted by clinical / biochemical deterioration, so even amongst the small number who were scanned but did not have a leak, most had another alternative complication (that likely stimulated a SIRs response) making them poor comparators.
Another question could be, is there an association of the postop CT findings to leak severity (ex. needing surgery/intervention vs conservative measures?
We agree this is an interesting clinical question and did explore it but no particular association was evident amongst this cohort. Unfortunately, our ability to answer this particular question was likely limited by the small sample.
Nevertheless, the study is well written, stats are sound, methodology could be ok except the drawback discussed above (non-comparison vs non-leak).
Thank you once again for your feedback.